# Analysis of Long-Term Water Level Variations in Qinghai Lake in China

**Jianmei Fang [1], Guijing Li [1], Matteo Rubinato [2], Guoqing Ma [3], Jinxing Zhou [1], Guodong Jia [1], Xinxiao Yu [1],\* and Henian Wang [4]**

[1] College of Soil and Water Conservation, Beijing Forestry University, Beijing 100083, China; jmf46@163.com (J.F.); lgj8023lhy@163.com (G.L.); zjx9277@126.com (J.Z.); jgd3@163.com (G.J.)

[2] School of Energy, Construction and Environment & Centre for Agroecology, Water and Resilience, Coventry University, Coventry CV1 5FB, UK; matteo.rubinato@coventry.ac.uk

[3] World Bank Loan Project Management Center of State Forestry and Grassland Administration, Beijing 100714, China; maguoqing8042@sina.com

[4] Institute of Wetland Research, Chinese Academy of Forestry, Beijing 100091, China; wanghenian2006@126.com

\* Correspondence: yuxinxiao111@126.com

**Abstract:** Qinghai Lake is the largest inland saline lake on the Tibetan Plateau. Climate change and catchment modifications induced by human activities are the main drivers playing a significant role in the dramatic variation of water levels in the lake ($\Delta h$); hence, it is crucial to provide a better understanding of the impacts caused by these phenomena. However, their respective contribution to and influence on water level variations in Qinghai Lake are still unclear and without characterizing them, targeted measures for a more efficient conservation and management of the lake cannot be implemented. In this paper, data monitored during the period 1960–2016 (e.g., meteorological and land use data) have been analyzed by applying multiple techniques to fill this gap and estimate the contribution of each parameter recorded to water level variations ($\Delta h$). Results obtained have demonstrated that the water level of Qinghai Lake declined between 1960 and 2004, and since then has risen continuously and gradually, due to the changes in evaporation rates, precipitation and consequently surface runoff associated with climate change effects and catchment modifications. The authors have also pinpointed that climate change is the main leading cause impacting the water level in Qinghai Lake because results demonstrated that 93.13% of water level variations can be attributable to it, while the catchment modifications are responsible for 6.87%. This is a very important outcome in the view of the fact that global warming clearly had a profound impact in this sensitive and responsive region, affecting hydrological processes in the largest inland lake of the Tibetan Plateau.

**Keywords:** climate change; water levels; causes and implications; Qinghai Lake, Tibetan Plateau

## 1. Introduction

The observed climate changes [1] had a significant impact on physical and natural processes on Earth during the past decade. The IPCC's (Intergovernmental Panel on Climate Change) latest report pointed out that if global warming will continue at its current rate, it could reach an increase in temperature up to 1.5 °C between 2030 and 2052 [2], causing the rising of sea levels as well as warming of water surfaces in oceans and lakes. Furthermore, human activities also had a strong impact on hydrological processes considering increased water consumption and situations of water shortage recorded around the world during the last decade. Hence, recent research focused on the impact of climate change and human activities on hydrological processes because this topic was identified by

many researchers as a priority [3,4] when planning for the future and making new developments more sustainable. Lake ecosystems usually provide indicators (i.e., water temperature, water levels, dissolved organic carbon (DOC)) of climate change, either directly or indirectly [5]. Recently, many studies have also been completed to investigate what has caused water level variations of lakes, such as the ones conducted on (i) the North American Great Lakes [6–8]; (ii) Lake Chad [9]; (iii) the Salton Sea [10] in the United States; (iv) Lake Lisan, Dead Sea rift [11]; and (v) Poyang Lake in China [12–14]. Summarizing the results achieved to date, hydrological conditions of each lake are affected by the lake's location, upstream boundaries, geographical climate and specific human activities undertaken on it such as residential developments, industry and irrigation tasks; hence, it is necessary to figure out the key factors that affect water levels to develop methods and procedures that can regulate those that alter or alleviate hydrological extreme processes in lakes.

The Tibetan Plateau (TP), known as "the roof of the world", "the third pole" and "the water tower of Asia", is the largest plateau in China and the highest plateau in the world [5,15], and it is considered the perfect location to identify the effects of global climate change [16–18]. Qinghai Lake is the largest inland saline lake on TP, and it has attracted extensive attention due to its special geographical location and its wide area characterized by fragile ecosystems. Over the last years, researchers have reported a drastic change in water levels in Qinghai Lake, indicating that the ecological environment around it is undergoing a rapid evolution [19–22]. Typically, inland closed lakes with no outlet streams are ideal to distinguish hydrologic processes and phenomena affecting the water balance because changes in water levels can result from limited factors such as precipitation, evaporation, groundwater infiltration [23–25] and the presence of specific vegetation [19].

The most effective way to estimate water levels in lakes is by applying the water-balance equation model, where the gain or loss of water directly reflects the changes in water levels [26,27]. Multiple inland lakes, due to the lack of funding and therefore resilient and accurate equipment to monitor basic data, are considered not appropriate to identify and quantify objectively the factors affecting the water balance. Nevertheless, Qinghai Lake, being a closed one with inlet river streams and without outlet river streams, is an ideal place to study, especially having the availability of meteorological and hydrological monitored data. To date, the study conducted by Li et al. [28] calculated the main water balance estimation of Qinghai Lake, while Cui et al. [20] preliminarily analyzed the climatic factors that affect the water level variations of Qinghai Lake. Despite this, there is still a need for new studies to fully distinguish and assess the relative contribution of anthropogenic activities and climate variability to water level variations in Qinghai Lake and their impacts on the corresponding water balance.

This paper present the analysis of water level variations recorded in Qinghai Lake during the last 57 years, examining the evolution and interpreting the impacts of driving factors to better understand the hydrological process of this inland lake basin in the northeast of TP, enhancing the present understanding of climatic variations on surface changes to provide a reference for local and regional water management.

The paper is organized as follows: Section 2 presents the description of the study area, introducing the data monitored and describing the statistical analysis used. Section 3 includes the estimation of long-term variations in Qinghai Lake as well as the impacts of climate change and catchment modifications for inflow runoff to Qinghai Lake. Section 4 provides a discussion of the results obtained, and Section 5 produces a brief summary and concluding remarks of the whole study.

## 2. Materials and Methods

### 2.1. Study Area and Data Availability

Qinghai Lake basin (97°50′~101°20′ E, 36°15′~38°20′ N) is located in the northeast of TP, covering an area of 29,664 km$^2$. The average annual air temperature ranges from −0.8 °C to 1.1 °C, and the average annual precipitation ranges from 327 to 423 mm. However, the annual precipitation is unevenly distributed, decreasing from the east and south to the west and north. The total surface runoff of

local main rivers including Buha River, Shaliu River, Haergai River, Heima River and Daotang River accounts for 83% of the total surface runoff into Qinghai Lake [28,29], with the first two (Buha & Shaliu) constituting the 64% of the surface runoff for the entire basin [30]. The main vegetation types in the basin are alpine meadows and alpine grasslands.

Qinghai Lake is the largest inland saline lake in this basin with an area of 4400 km$^2$ (in 2016), and it is located at an altitude of 3193 m above sea level. As previously mentioned in Section 1, this lake is a closed one with no surface water outflow. It is about 106 km in length from east to west, and 63 km in width from north to south, and 360 km in circumference [30]. The average annual air temperature above the lake is about 1.2 °C, and the average annual precipitation near its proximity is about 357 mm.

The datasets available used for this study can be summarized as follows:

- Daily water levels in Qinghai Lake at Xiashe station (36°35′ N, 100°29′ E) from 1959 to 2016, obtained by the Information Center of Qinghai Hydrographic Bureau, China (ICQHB).
- Daily surface runoff of Buha River and Shaliu River, observed at the estuary of Buha River station (37°18′ N, 99°44′ E, from 1960 to 2016), at Gangcha station (37°17′ N, 100°19′ E, from 1960 to 1975) and at Gangcha II station (36°19′ N, 100°18′ E, from 1976 to 2016, obtained as well by ICQHB.
- Meteorological data:

   i.　Daily meteorological data of 14 national meteorological stations from 1960 to 2016, obtained by the China Meteorological Information Center.
   ii.　Monthly meteorological data from 1960 to 2010 at three meteorological stations, obtained by Qinghai Meteorological Bureau in China.
   iii.　Daily precipitation data of Buha River rain station from 1962 to 2016 obtained by ICQHB.
   iv.　Daily evaporation data from 1984 to 2016 at Xiashe station obtained from ICQHB.
   v.　Yearly evaporation data from 1960 to 1988 obtained from the literature [29]. (Figure 1 and Table 1).

- Environmental and physical details of Qinghai Lake, and these datasets were obtained from ICQHB and the literature [30].
- Land use data from 1980 to 2015, obtained by the Data Center of Resources and Environmental Sciences, Chinese Academy of Sciences.

**Table 1.** Detailed information of the meteorological stations in and around Qinghai Lake basin used to collect the datasets previously described.

| No. | Station Number | Station Name | Latitude (°N) | Longitude (°E) | ASL (m) | Data Collection Frame |
|---|---|---|---|---|---|---|
| 1 | 52,645 | Yeniugou | 38.43 | 99.60 | 3315 | 1960–2016 |
| 2 | 52,842 | Chaka | 36.78 | 99.08 | 3088 | 1960–2016 |
| 3 | 52,633 | Tuole | 38.82 | 98.42 | 3368 | 1960–2016 |
| 4 | 52,833 | Wulan | 36.93 | 98.48 | 2951 | 1960–2016 |
| 5 | 52,836 | Dulan | 36.30 | 98.10 | 3190 | 1960–2016 |
| 6 | 52,737 | Delingha | 37.37 | 97.38 | 2982 | 1960–2016 |
| 7 | 52,868 | Guide | 36.02 | 101.37 | 2274 | 1960–2016 |
| 8 | 52,657 | Qilian | 38.18 | 100.25 | 2788 | 1960–2016 |
| 9 | 52,754 | Gangcha | 37.33 | 100.13 | 3302 | 1960–2016 |
| 10 | 52,856 | Gonghe | 36.27 | 100.62 | 2836 | 1960–2016 |
| 11 | 52,943 | Xinghai | 35.58 | 99.98 | 3324 | 1960–2016 |
| 12 | 52,765 | Menyuan | 37.38 | 101.62 | 2851 | 1960–2016 |
| 13 | 52,866 | Xining | 36.73 | 101.75 | 2296 | 1960–2016 |
| 14 | 52,955 | Guinan | 35.58 | 100.73 | 3121 | 1960–2016 |
| 15 | 52,745 | Tianjun | 37.30 | 99.02 | 3417 | 1961–2010 |
| 16 | 52,855 | Huangyuan | 36.68 | 101.25 | 2675 | 1961–2010 |
| 17 | 52,853 | Haiyan | 36.90 | 100.98 | 3010 | 1961–2010 |
| 18 | 1,329,500 | The estuary of Buha River | 37.03 | 99.73 | 3191 | 1962–2016 |

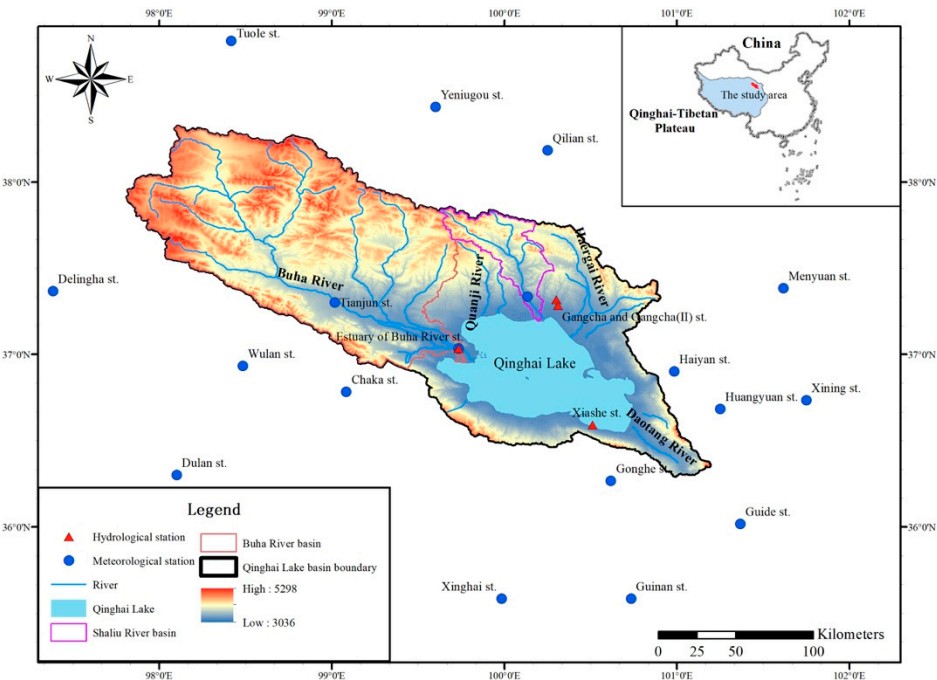

**Figure 1.** The location of Qinghai Lake basin in China (top right corner), physical characteristics of Qinghai Lake in Qinghai Lake basin (centre) and the spatial distribution of the hydrological stations (red triangles) and meteorological stations (blue dots).

### 2.2. Governing Equations

#### 2.2.1. Lake Water Balance Model

As Qinghai Lake is a closed-catchment with no surface water outflow, the annual hydrological water balance equation can be expressed as follows:

$$\Delta h = P_l - E_l + R_{ls} + R_{lg} \pm \varepsilon \tag{1}$$

where $\Delta h$ is the yearly water level variation, (mm); $P_l$ is the yearly precipitation on the lake surface, (mm); $E_l$ is the yearly evaporation from the lake surface, (mm); $R_{ls}$ is the yearly surface runoff into the lake, (mm); $R_{lg}$ is yearly underground runoff on the lake bottom, (mm); $\varepsilon$ is the error, (mm). For this watershed scenario, the surface runoff is almost equivalent to the river runoff and the slope surface runoff can be considered negligible. $\Delta h$ can be quantified as well as follows:

$$\Delta h = h_i - h_{i-1} \tag{2}$$

where $h_i$ and $h_{i-1}$ are the lake level at $i$ year and at $i - 1$ year.

The yearly average water level of the lake was obtained from the daily water level data at Xiasha station. $P_l$ was calculated by applying the Thiessen Polygon method focusing in the area between Buha River station, Gangcha station, Haiyan station and Gonghe station, which are the nearest stations to the lake. $E_l$ was obtained from evaporation pan (type of E20) data at Xiashe station [29]. The yearly total surface runoff ($R_{ls}$, mm) into the lake was obtained from the surface runoff ($Q_{ls}$, m³) of Buha River and Shaliu River by using the proportion amplification method [31].

In this paper, $\Delta h$ is subject to a combination of climate and human activities effects (such as farmland reclamation, grazing, afforestation which indirectly influence runoff and catchment characteristics). $P_l$ and $E_l$ represents the climate variability, $R_{ls}$ and $R_{lg}$ are the results of the combination of climate and catchment modifications. Hence, to correctly quantify the contribution rate of climate and catchment characteristics (human induced) to $\Delta h$, it is necessary to calculate accurately the contribution rate

of climate and catchment characteristics (human induced) to $R_{ls}$ and $R_{lg}$. Therefore, the calculation method can be applied as follows:

$$R(\Delta h)_c = R(\Delta h_{Pl})_c + R(\Delta h_{El})_c + R(\Delta h_{Rls})_c + R(\Delta h_{R\lg \pm \varepsilon})_c \tag{3}$$

$$R(\Delta h)_u = R(\Delta h_{Rls})_u + R(\Delta h_{R\lg \pm \varepsilon})_u \tag{4}$$

where $R(\Delta h)_c$, $R(\Delta h)_u$ respectively represent the contribution rate of climate and catchment modifications to $\Delta h$. $R(\Delta h_{Pl})_c$, $R(\Delta h_{El})_c$ respectively represent the climate change contribution of $P_l$ and $E_l$ to $\Delta h$. $R(\Delta h_{Rls})_c$, $R(\Delta h_{Rls})_u$ respectively represent the $R_{ls}$ contribution rate of climate change and catchment change to $\Delta h$. $R(\Delta h_{R\lg \pm \varepsilon})_c$, $R(\Delta h_{R\lg \pm \varepsilon})_u$ respectively represent the $R_{lg} \pm \varepsilon$ contribution rate of climate change and catchment change to $\Delta h$.

### 2.2.2. Land Use Dynamic Index

Land use change can reflect the effect as well as the intensity of human activities. The Land Use Dynamic Index was proposed by Chen et al. [32] and was adopted in this study to describe the change of land use types in the research area for a certain period (1980–2015). The calculation method was completed as follows:

$$LC = \frac{\sum\limits_{i=1}^{n} \Delta U_{i\text{in}}}{2 \sum\limits_{i=1}^{n} U_{i0}} \times \frac{1}{T} \times 100\% \tag{5}$$

where $LC$ is the Land Use Dynamic Index in a certain period of time in the research area (%), $\Delta U_{i\text{in}}$ refers to the area of type $i$ land use converted into the non-$i$ type land use within a certain period of time in the research area (km$^2$), $U_{i0}$ is the area of type $i$ land use at the beginning of the study period (km$^2$), $T$ is the research period (years).

### 2.2.3. Statistical Analysis

(1) The Non-parametric Mann-Kendall Test

The non-parametric Mann-Kendall test [33,34] (M-K test) and the cumulative anomaly method were used to detect any point of abrupt changes in the variables considered. The M-K test has been widely applied to identify the point at which the hydrological processes change significantly due to the climate [35,36]. The details about this statistical method can be obtained in the relevant literature [37].

First, the partial M-K test statistics are calculated as:

$$S_k = \sum_{i=1}^{k} \sum_{j=1}^{i-1} \alpha_{ij} \quad (k = 2, 3, 4, \ldots, n) \tag{6}$$

$$\alpha_{ij} = \begin{cases} 1 & x_i > x_j \\ 0 & x_i \le x_j \end{cases} \quad 1 \le j \le i \tag{7}$$

Statistical variable $UF$ is adopted and defined as:

$$UF = \frac{S_k - E(S_k)}{\sqrt{Var(S_k)}} \quad (k = 1, 2, 3, \ldots, n) \tag{8}$$

$$E(S_k) = \frac{k(k-1)}{4} \tag{9}$$

$$Var(S_k) = \frac{k(k-1)(2k+5)}{72} \tag{10}$$

Proceed to Equation (11) putting the data sequence $x$ in reverse order:

$$\begin{cases} UB_k = -UF_{k'} \\ k' = n+1-k \end{cases} \quad (k = 1, 2, 3, \ldots, n) \tag{11}$$

In the M-K curve, if the value of the intersection of the curve forward ($UF$) or the curve backward ($UB$) is greater than 0, this suggests that the record sequence shows an upward trend; less than 0 suggests a downward trend. When the record exceeds the critical line (Given the significance level $\alpha = 0.05$, the critical lines $U_{0.05} = \pm 1.96$), this suggests that an increase or decrease in the trend may be significant. The range of exceeding the critical line is the time zone in which the abrupt change occurs. If there is an intersection between the curves of $UF$ and $UB$ in the range of the critical lines, the time of the intersection is the time of the abrupt change started [38].

(2) The Cumulative Anomaly Method

The cumulative anomaly method is widely used to indicate the runoff [39], precipitation and other factors that deviate from the normal situations, focusing on the difference between a certain value and the average value of a series [38].

(3) The Principal Component Regression Analysis

The principal component regression (PCR) analysis [40] is a combination of principal component analysis and regression analysis. Typically, this method considers regressing the outcome on a set of covariates based on a standard linear regression model, using PCA (principal component analysis) for estimating the unknown regression coefficients. Generally, only a subset of all the principal components for regression is used; hence, PCR tends to act as a regularized procedure.

(4) The Grey Relational Analysis

The grey relational analysis [41] is adopted in this study to solve uncertain problems such as limited data and incomplete information by calculating the grey correlation degree $\gamma_i$, quantifying the correlation degree among the influential factors of underground runoff.

(5) The Least Square Method

The least square method [42] is applied to procure unknown data and minimize the sum of squared errors between the obtained data and the actual data. The least square method can also be used for curve fitting.

(6) The Partial Least Squares Regression Method

The partial least squares regression method [43] is a combination of multiple linear regression analysis, canonical correlation analysis and principal component analysis, reflecting the influence of the sample population on the predicted values and fully considering the influence of the comprehensive effect between individual factors on the predicted ones.

2.2.4. Sensitivity Analysis Based on the Budyko Framework

Climate change and human activities are the most important drivers to determine the river hydrological process of the catchment [44]. In this study, the sensitivity coefficient method [45] based on Budyko Theory [46] was used to quantitatively separate the impacts of climate change and human activities on the variations of streamflow into Qinghai Lake. The theoretical equation of Budyko curve [47] can be applied as follows:

$$\frac{ET}{P} = 1 + \frac{ET_0}{P} - \left[1 + \left(\frac{ET_0}{P}\right)^{\omega}\right]^{1/\omega} \tag{12}$$

where *ET* is the evapotranspiration of the upper catchment area, (mm); *P* is the precipitation of the catchment area, (mm); $ET_0$ is the potential evapotranspiration of the catchment area, (mm); the empirical parameter $\omega$ represent catchment characteristics, such as human activities, land use, vegetation, topography, and properties of soil [48,49], [$\omega \in (1, \infty)$].

The change of surface runoff in a given basin can be characterized by climate and human activities changes as follows:

$$\Delta Q = \Delta Qc + \Delta Qu \tag{13}$$

where $\Delta Qc$ and $\Delta Qu$ represent the surface runoff variation caused by climate change and human activities changes, respectively. The surface runoff variation caused by climate change can be expressed by the following formula [45]:

$$\Delta Qc = \frac{\partial Q}{\partial P} \times \Delta P + \frac{\partial Q}{\partial ET_0} \times \Delta ET_0 \tag{14}$$

The surface runoff variation caused by human activities can be expressed by the following formula:

$$\Delta Qu = \frac{\partial Q}{\partial \omega} \times \Delta \omega \tag{15}$$

where $\Delta P$ is the variation of precipitation, $\Delta \omega$ is the variation of the empirical parameter $\omega$ of a given catchment; $\Delta ET_0$ is the potential evapotranspiration variation; $\frac{\partial Q}{\partial P}$, $\frac{\partial Q}{\partial ET_0}$, $\frac{\partial Q}{\partial \omega}$ respectively represent the sensitivity coefficient of runoff to precipitation, runoff to potential evapotranspiration, runoff to precipitation, runoff to the empirical parameter represent catchment characteristics. All of the sensitivity coefficients can be calculated as follows:

$$\frac{\partial Q}{\partial P} = \left[1 + \left(\frac{ET_0}{P}\right)^\omega\right]^{(1/\omega - 1)} \tag{16}$$

$$\frac{\partial Q}{\partial ET_0} = \left[1 + \left(\frac{P}{ET_0}\right)^\omega\right]^{(1/\omega - 1)} - 1 \tag{17}$$

$$\frac{\partial Q}{\partial \omega} = [P^\omega + ET_0{}^\omega]^{1/\omega} \cdot \left[\left(-\frac{1}{\omega^2}\right) \cdot \ln(P^\omega + ET_0{}^\omega) + \frac{1}{\omega} \cdot \frac{1}{P^\omega + ET_0{}^\omega} \cdot (\ln P \cdot P^\omega + \ln ET_0 \cdot ET_0{}^\omega)\right] \tag{18}$$

In this paper, the potential evapotranspiration at the meteorological stations was calculated by applying the FAO56 method, Penman-Monteith model [50], because previous literature [51,52] has demonstrated how these methods are reliable to estimate potential effects of climate change on the calculation of the evaporation as well as the influence of climate change on water cycles. *P* and $ET_0$ of the entire basin were obtained by applying the area-weight method of Tyson Polygon.

## 3. Results and Analysis

### 3.1. Long-Term Variations in Water Levels and the Hydro-Climatic Factors

#### 3.1.1. Long-Term Variations in Water Levels

Figure 2 shows the annual water levels of Qinghai Lake recorded during the period 1960–2016. Overall, comparing datasets within other locations in the semi-arid areas of Western China, the water level varied significantly ($\approx$3.5 m) and it is possible to notice a clear inflection point recorded in 2004. Herein, the analysis of the graph was divided into two periods to simplify the procedure: period I was selected to be between 1960 and 2004, while period II was selected to be between 2005 and 2016. The annual water level of the lake declined at the rate of 7.84 cm/year ($P < 0.001$), with a total decrease of 3.46 m in period I, while the annual water level of the lake has risen at the rate of 13.80 cm/year ($P < 0.001$), with a total increase of 1.49 m in period II.

The variation of the water levels of the lake ($\Delta h$) reflected the acquisition and loss of water volume over the years due to multiple factors, and the trend is shown in Figure 3. According to the results obtained, $\Delta h$ tended to increase during period I ($R^2 = 0.0105$, $P < 0.01$) as well as during period II ($R^2 = 0.0291$, $P < 0.01$). The increasing rate of the water level in period II was notably faster than the one in period I, and the water level of the lake in 1960 could be reached again by 2030 if the present increasing rate continues constantly.

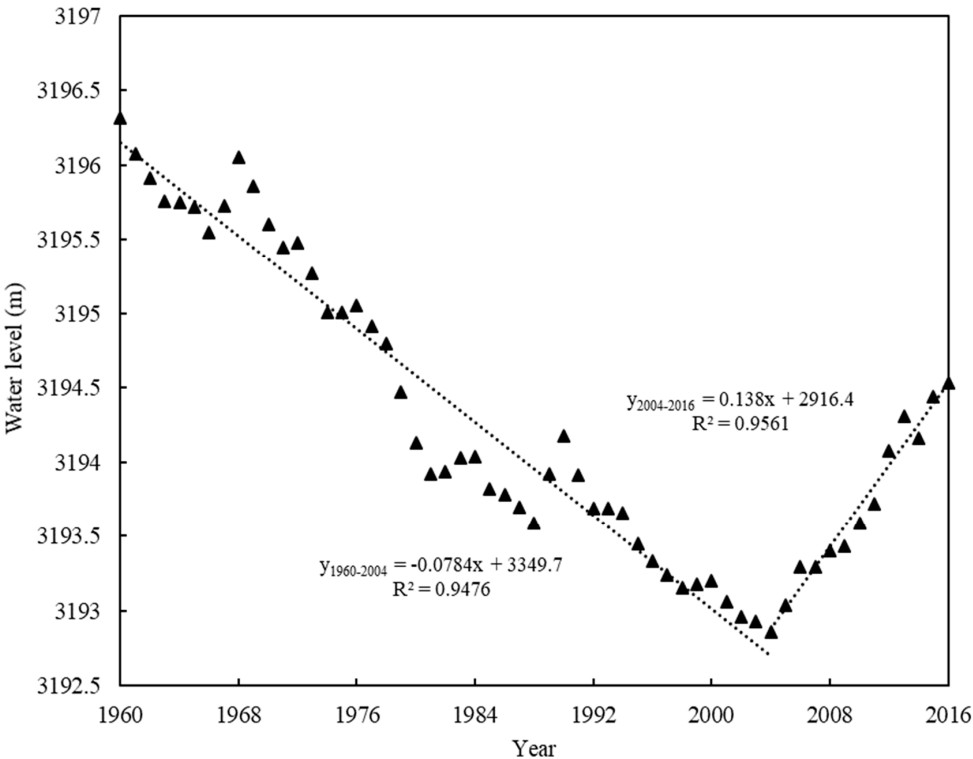

**Figure 2.** The annual water levels of Qinghai Lake recorded during the period 1960–2016.

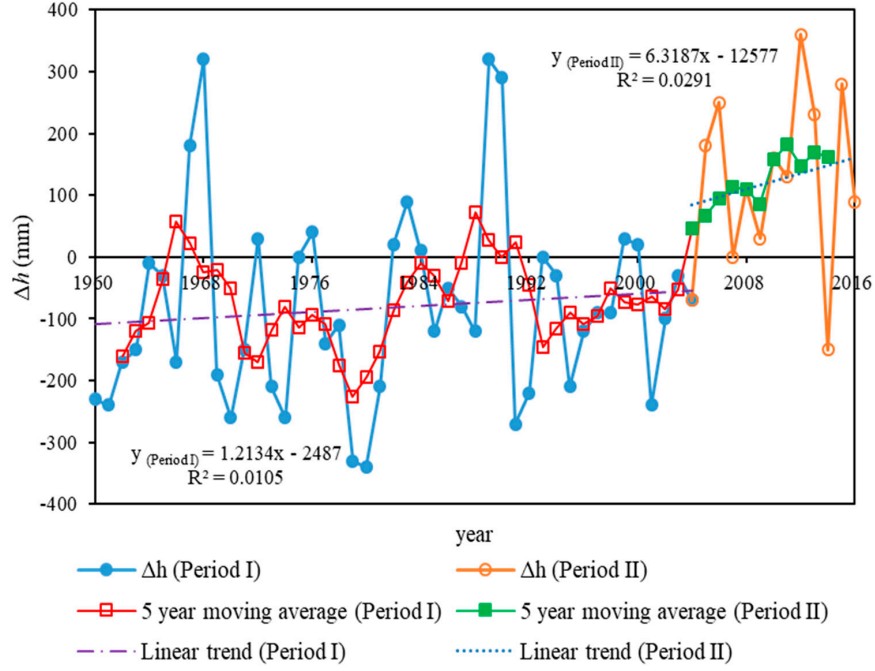

**Figure 3.** The trend of water level variation ($\Delta h$) during the period of 1960–2016.

### 3.1.2. Analysis of Hydro-Climatic Factors Influencing Water Levels

$\Delta h$ was dependent on the lake hydro-climatic conditions $P_l$, $E_l$, $R_{ls}$ and $R_{lg} \pm \varepsilon$. All together, these variables affected the rise or fall of the water level in the lake, and the relationship between them and the corresponding variations in the water levels of the lake are shown in Figure 4 and Table 2.

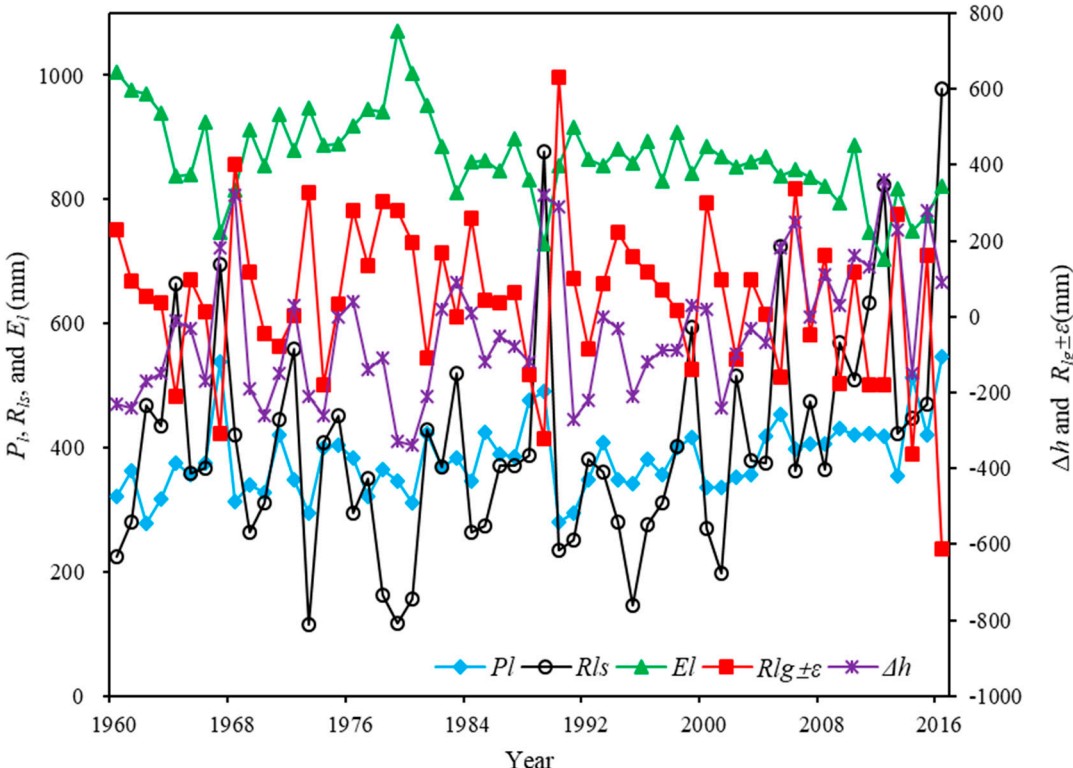

**Figure 4.** Variations of the hydro-climatic factors of Qinghai Lake during the period of 1960–2016. $P_l$ is the precipitation over the lake (mm), $E_l$ is the evaporation from the lake surface (mm), $R_{ls}$ is the surface runoff into the lake (mm), $R_{lg} \pm \varepsilon$ is the underground runoff and study error (mm), $\Delta h$ is lake level variation (mm).

**Table 2.** The hydro-climatic factors calculated in different periods (mm) *.

| Periods | $P_l$ | $R_{ls}$ | $R_{lg} \pm \varepsilon$ | $E_l$ | $\Delta h$ |
|---------|-------|----------|-------------------------|-------|-----------|
| I (1960–2004) | 367.94 (+45.67%) | 364.02 (+45.18%) | 73.68 (+9.15%) | 887.64 (−100%) | −82.00 |
| II (2005–2016) | 432.77 (+43.38%) | 564.93 (+56.62%) | −55.81 (−6.50%) | 802.73 (−93.50%) | +139.17 |
| 1960–2016 | 381.59 (+45.74%) | 406.32 (+48.70%) | 46.42 (+5.56%) | 869.77 (−100.00%) | −35.44 |

\* The bracketed values refer to the percentage of total input or output represented by average yearly volumetric flux (mm) changes at different periods. Sign + represents water in, while sign - represents water out.

During the period under investigation, surface runoff into the lake ($R_{ls}$) was mainly due to the Buha River and Shaliu River, which contributed about 48.70% of the total water in the lake. The underground runoff on the lake bed and the associated error ($R_{lg} \pm \varepsilon$) accounted for 5.56% of the total water intake of the lake, and this value fluctuated significantly. The precipitation on the lake surface ($P_l$) contributed about 45.74% of the total water in the lake while the evaporation from the lake surface ($E_l$) contributed about 100% of the total water removed from the lake. As possible to notice from Figure 4, $P_l$, and $R_{ls}$ had similar trends, while this could not be confirmed for $E_l$ and $R_{lg}$. The peak of $\Delta h$ often corresponded to the peak of $P_l$, $R_{ls}$.

The annual average values for each hydro-climatic variable $P_l$, $R_{ls}$, $R_{lg} \pm \varepsilon$, and $E_l$ were 381.59 (mm), 406.32 (mm), 46.42 (mm), and 869.77 (mm), respectively. In period I, approximately 45.67% of the total water input into the lake came via $P_l$, with 45.18% of water input coming from $R_{ls}$, and a small

fraction of water input was due to $R_{lg} \pm \varepsilon$. The whole outflow was estimated to be associated with $E_l$. In period II, approximately 43.38% and 56.62% of the total water input was associated with $P_l$ and $R_{ls}$, respectively, while $E_l$ contributed 93.5% to the outflow with a small fraction of water escaping the lake attributed to $R_{lg} \pm \varepsilon$ (6.5%). This indicated that the water balance of Qinghai Lake was mainly determined by $P_l$, $R_{ls}$ and $E_l$. Therefore, the authors can confirm that $R_{lg} \pm \varepsilon$, always being a small percentage, accounted for a small proportion of the water balance of Qinghai Lake.

### 3.2. Causes of Changes in Water Levels of the Lake

#### 3.2.1. Impact of Climate Change on Water Levels

Figure 4 shows the relationship between $P_l$, $E_l$ and $\Delta h$. The correlation coefficient between $P_l$ and $\Delta h$ calculated was 0.356 ($P < 0.01$). The fluctuation range for $P_l$ was estimated between 277.2 and 546.4 (mm), where the mean value was obtained equal to 381.6 (mm), showing an upward trend at the rate of 1.4347 mm/year ($R^2 = 0.164$, $P < 0.01$). The correlation coefficient between $E_l$ and $\Delta h$ was $-0.705$ ($P < 0.01$) and the fluctuation range for $E_l$ was between 702.6 and 1070.5 (mm), where the mean value was 869.8 (mm), showing a downward trend at the rate of 2.2823 mm/year ($R^2 = 0.290$, $P < 0.01$). Changes in behavior for $P_l$ were consistent with the fluctuations of $\Delta h$, and the peaks of $\Delta h$ were noticed to be delayed by 1 year when comparing them with those associated with $P_l$. However, $E_l$ was generally contrary to the fluctuations of $\Delta h$.

Based on the results achieved, $E_l$ and $P_l$ were identified as the main important climate factors affecting the water level changes, and the correlation relationships between $E_l$ and $\Delta h$ estimated were of higher quality than those obtained between $P_l$ and $\Delta h$.

It was clear that at peaks in the rising periods for $\Delta h$ corresponded to higher values of $P_l$ but lower values of $E_l$ (e.g., 1968, 1989, and 2012). Furthermore, decreasing peaks of $\Delta h$ corresponded to lower values of $P_l$ but higher values of $E_l$ (e.g., 1980 and 1991). Precipitation rates were quantified to affect both the runoff of the inflow rivers and underground runoff acting on the water level changes. Finally, evaporation was selected as the only factor of climate influencing water "exiting" the lake, playing a significant role in the fluctuation of the water level.

#### 3.2.2. Impact of Human Activities on Catchment Modifications and Consequently on Water Levels

Figure 5 illustrates 7 different years spanning 1980 and 2015, to highlight the land use changes from 1980 to 2015 obtained by the superposition function fitted within ArcGIS10.2. Despite being present and noticeable, land use changes observed were not particularly significant as possible to notice in Figure 5 (Tables 3 and 4).

**Table 3.** The land use dynamic attitude (LC) from 1980 to 2015.

| Period | Period | | | | | |
|---|---|---|---|---|---|---|
| | 1980–1990 | 1190–1995 | 1995–2000 | 2000–2005 | 2005–2010 | 2010–2015 |
| LC (%) | 0.06 | 0.04 | 0.05 | 0.03 | 0.01 | 0.03 |

**Table 4.** Land use change parameters from 1980 to 2015 (km$^2$).

| Type | Farmland | Forestland | Grassland | Water Area | Constructive Land | Unused Land | Total |
|---|---|---|---|---|---|---|---|
| Farmland | 493 | / | 1 | 1 | 4 | 1 | 500 |
| Forestland | / | 1371 | 10 | 2 | 1 | 1 | 1385 |
| Grassland | 64 | 5 | 17,475 | 30 | 7 | 16 | 17,596 |
| Water area | / | / | 142 | 4842 | / | 157 | 5141 |
| Constructive land | / | / | / | / | 26 | / | 26 |
| Unused land | / | / | 15 | 27 | / | 4975 | 5017 |
| Total | 557 | 1376 | 17,643 | 4901 | 37 | 5149 | 29,664 |

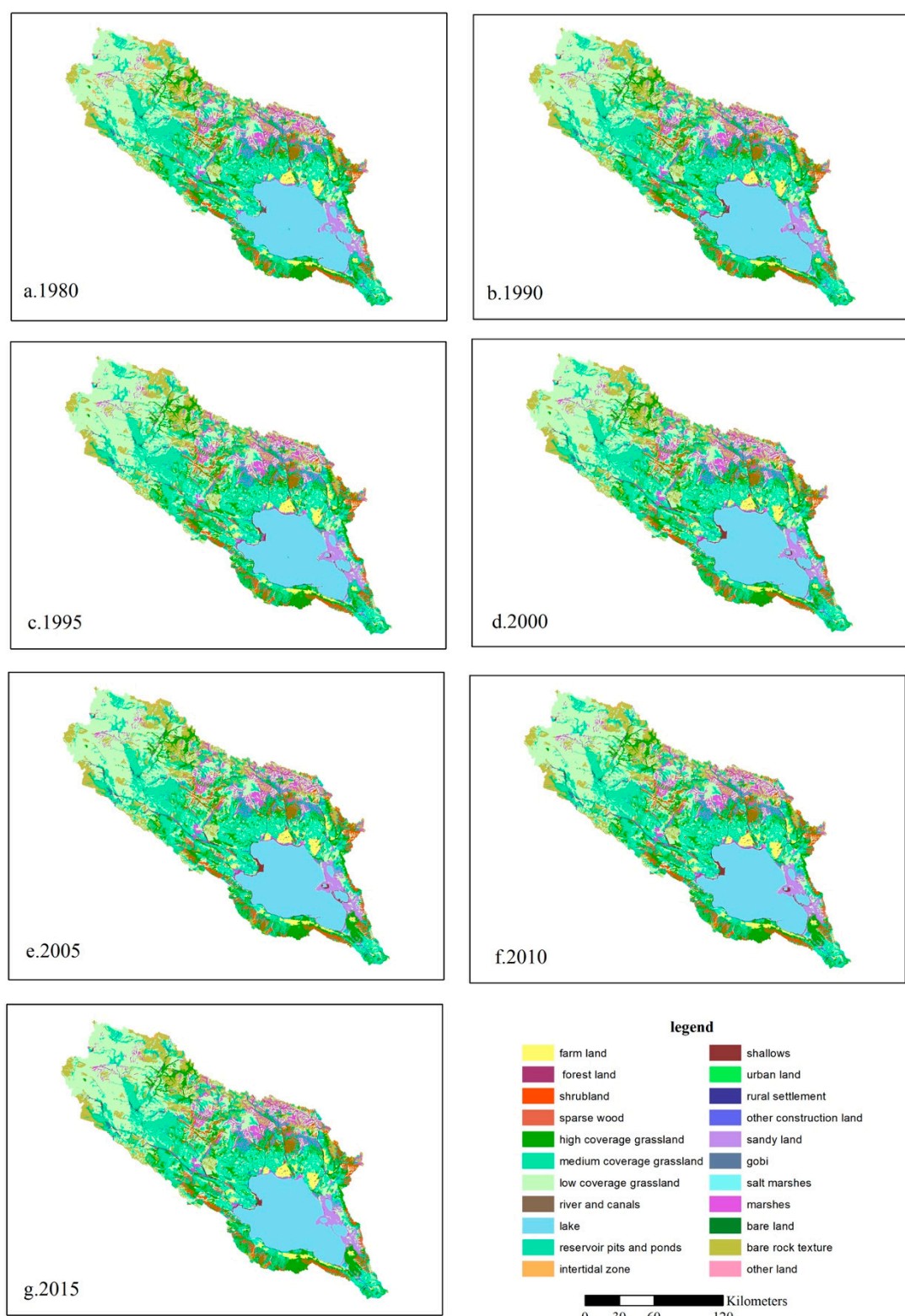

**Figure 5.** The changes in land use in Qinghai Lake basin from 1980 to 2015.

As the overall topography of Qinghai Lake basin is relatively gentle, the production and confluence of the runoff typically have longer durations, highly dependent on the catchment. The empirical parameter representing land surface characteristics of the basin ($\omega$) was calculated by applying the least square method [45] according to Equation (6) and results are displayed in Figure 6. The correlation coefficient between $\omega$ and $\Delta h$ was $-0.262$ ($P < 0.05$). Figure 6 and Table 4 confirmed as previously

stated that there were little changes in the land use, and the main reasons causing $\omega$ changes were not associated with the land change use, but were probably due to changes in local vegetation and soil conditions.

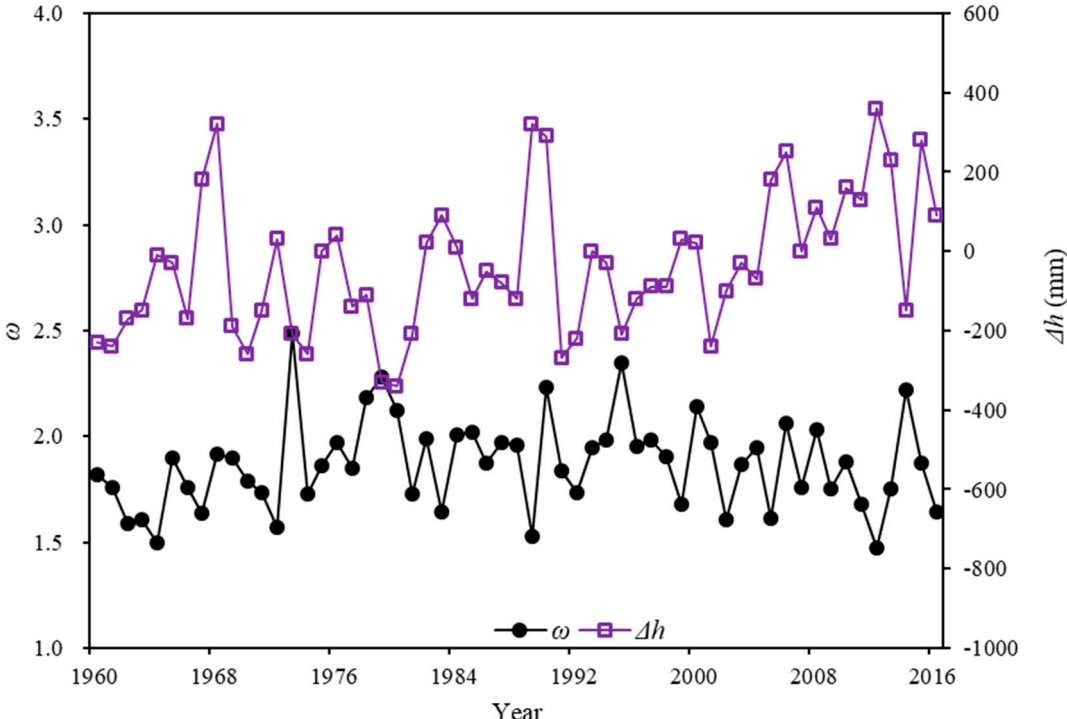

**Figure 6.** Variation the empirical parameter representing catchment characteristics ($\omega$) and $\Delta h$ in Qinghai Lake basin during the period of 1960–2016.

By summarizing all the contributions, results have also confirmed that affecting factors $\Delta h$, $P_l$ and $E_l$ were entirely attributable to climate change, while $R_{ls}$ and $R_{lg} \pm \varepsilon$ were the results of the joint action between climate and catchment modifications. Therefore, it became necessary to focus on $R_{ls}$ and $R_{lg} \pm \varepsilon$ and to quantify their impact due to climate change and catchment change.

3.2.3. Impact of Climate and Catchment Modifications on the Surface Runoff ($R_{ls}$)

Surface runoff is a crucial variable to consider when completing any lake water balance [20,53] and in this study it accounted for 45.18~56.62% of the total lake inflow during the study period, which demonstrates how this parameter was a key factor affecting the water level variations in the lake. The correlation coefficient between $R_{ls}$, and $\Delta h$ was calculated to be 0.590 ($P < 0.01$).

Surface runoff was generated from the surrounding catchment area of about 25,000 km$^2$ (obtained by subtracting the lake area from the total basin area). Surface runoff (mm) was obtained by dividing the annual total runoff of the basin (m$^3$) by the annual catchment area. The catchment area of annual maximum, annual minimum and mean annual values was 25,439.71 km$^2$ (in 2004), 25,136.70 km$^2$ (in 1960), and 25,308.56 km$^2$, respectively. In this paper, the Mann-Kendall test method (M-K test) and the cumulative anomaly method were used to identify remarkable changes in the variables' behaviors (i.e., surface runoff, precipitation and potential evapotranspiration). As shown in Figure 7a, the precipitation in the basin is always rising ($UF > 0$), and the precipitation trend shows a significant change from 2005 ($UF > 1.96$). The intersection of $UF$ and $UB$ curves indicates an abrupt change point in 2003. Furthermore, when focusing on the evapotranspiration trends in Figure 7b, and intersection point was noticed between the two curves in 1998. The M-K test failed to identify any abrupt change point in the trend of the surface runoff recorded, while the cumulative anomaly analysis method correctly estimated it as observed in 2004 (Figure 7c). The results indicated that surface runoff, precipitation and lake

water levels were closely related. Additionally, the year corresponding to the anomalies noted in the variation of the lake water levels and the surface runoff slightly lagged behind the year corresponding to the abrupt change point related to the precipitation. Based on the cumulative anomaly analysis method, the runoff series were divided into two periods like the variation of water levels: period I (1960–2004) and period II (2005–2016), which enabled the authors to calculate the basin characteristic parameters and sensitivity coefficients for period I and period II that are presented in Table 5.

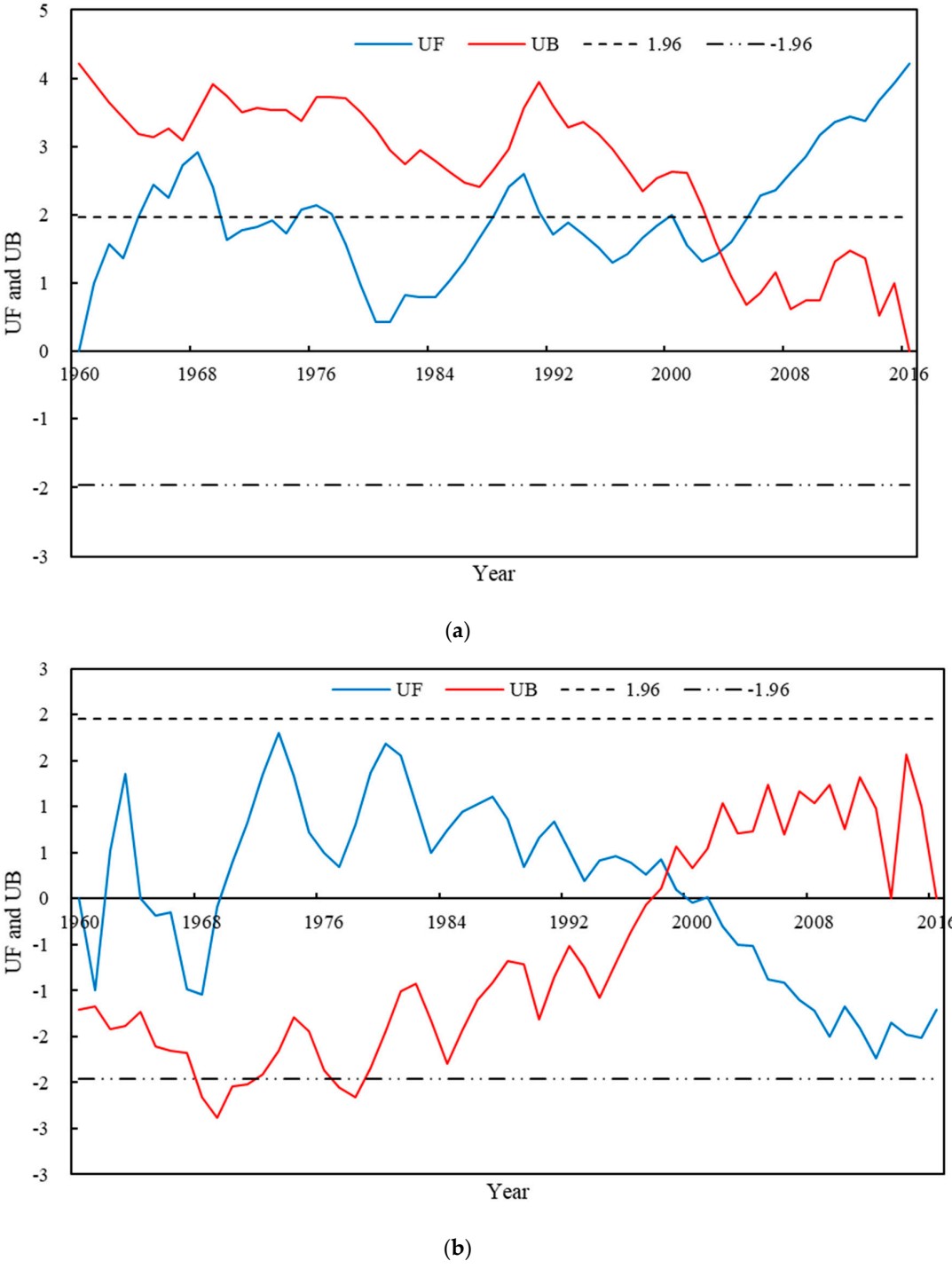

**Figure 7.** *Cont.*

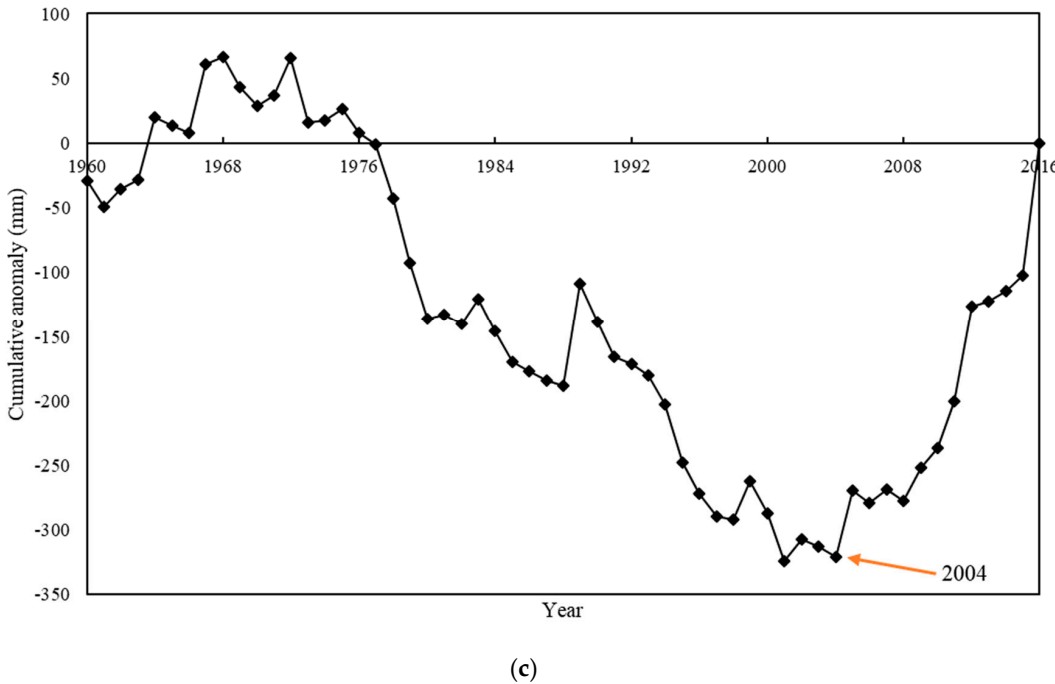

(**c**)

**Figure 7.** Mann-Kendall test of annual precipitation from 1960 to 2016 (**a**), Mann-Kendall test of annual potential evapotranspiration (**b**), and Cumulative anomaly of the annual surface runoff (**c**) in Qinghai Lake basin.

**Table 5.** Basin characteristic parameters and sensitivity coefficients in each study period.

| Variable | Study Period | | |
|---|---|---|---|
|  | 1960 to 2016 | I | II |
| $Q$ (mm) | 69.96 | 62.84 | 96.69 |
| $P$ (mm) | 349.13 | 334.65 | 403.42 |
| $ET_0$ (mm) | 1078.55 | 1081.23 | 1068.49 |
| $\omega$ | 1.85 | 1.87 | 1.82 |
| $\partial Q/\partial P$ | 0.36 | 0.34 | 0.42 |
| $\partial Q/\partial ET_0$ | −0.05 | −0.05 | −0.07 |
| $\partial Q/\partial \omega$ | −115.75 | −106.43 | −146.57 |

According to the results, the total surface runoff variation was measured as +33.9 mm. Contribution rates of climate change and catchment modifications to this variation were calculated using Equations (14) and (15) and the results show that the main cause of runoff change from 1960 to 2016 was climate change (producing an increased surface runoff by 25.54 mm for a corresponding contribution rate of 80.19%). On the other hand, the effect caused by catchment modifications was not to be considered a negligible factor, considering that it generated an increase in surface runoff of 6.31 mm and hence its contribution rate was 19.81%. Error was estimated to be 2 mm, corresponding to 5.91% of the total variation (Table 6).

**Table 6.** The results of attribution analysis of runoff change.

|  | $\Delta Q$ | $\Delta Qc$ | $\Delta Qu$ | Error |
|---|---|---|---|---|
| Contribution Amount (mm) | 33.85 | 25.54 | 6.31 | 2.00 |
| Contribution Rate (%) | 100 | 80.19 | 19.81 | 5.91 |

3.2.4. Impact of Climate and Catchment Modifications on the Underground Runoff ($R_{lg} \pm \varepsilon$)

$R_{lg} \pm \varepsilon$ was included in the water balance equation model but due to limitations in data availability, it was more challenging to accurately estimate the effects on its variations due to climate and catchment

modifications. However, by applying the equations described in Section 2, the correlation between $R_{lg} \pm \varepsilon$ and $\Delta h$ was calculated to be 0.143.

Underground runoff could be affected directly and indirectly by multiple factors; therefore, the authors believed it was not appropriate to complete a comprehensive and systematic analysis of its dynamic changes by using a single factor analysis. Hence, the principal component regression method [40] was used. The factors $x_i$ identified to influence the underground runoff ($y$) can be summarized as follows:

- water level variation in the lake ($x_1$);
- precipitation on the lake surface area ($x_2$);
- surface runoff of the basin ($x_3$);
- evaporation from the lake surface ($x_4$);
- precipitation across the entire basin area ($x_5$);
- empirical parameter representing land surface characteristics of the basin ($x_6$).

The correlation analysis is shown in Table 7; the multi-collinearity among the influential factors, and the grey relational degree analysis (Table 8) show that each factor had a closed relationship with $y$, and the grey relational degree of each factor $x_i$ is displayed as $\gamma_i$.

**Table 7.** Correlation coefficient matrix of influential factors.

| Correlation Coefficient | $x_1$ | $x_2$ | $x_3$ | $x_4$ | $x_5$ | $x_6$ |
|---|---|---|---|---|---|---|
| $x_1$ | 1 | 0.36 ** | 0.51 ** | −0.71 ** | 0.39 ** | −0.26 * |
| $x_2$ | 0.36 ** | 1 | 0.61 ** | −0.56 ** | 0.97 ** | −0.28 * |
| $x_3$ | 0.51 ** | 0.61 ** | 1 | −0.56 ** | 0.64 ** | −0.59 ** |
| $x_4$ | −0.71 ** | −0.56 ** | −0.56 ** | 1 | −0.55 ** | 0.31 * |
| $x_5$ | 0.39 ** | 0.97 ** | 0.64 ** | −0.55 ** | 1 | −0.35 ** |
| $x_6$ | −0.26 * | −0.28 * | −0.59 ** | 0.31 * | −0.35 ** | 1 |

$* P < 0.05, ** P < 0.01.$

**Table 8.** Grey relational degree matrix of influential factors on subsurface runoff.

| Incidence Matrix | $\gamma_1$ | $\gamma_2$ | $\gamma_3$ | $\gamma_4$ | $\gamma_5$ | $\gamma_6$ |
|---|---|---|---|---|---|---|
| $y$ | 0.8205 | 0.7683 | 0.7739 | 0.8295 | 0.7609 | 0.8441 |

It can be seen from Table 9 that the cumulative contribution rates of the first, second and third principal components ($F_1$, $F_2$, $F_3$) were more than 80%, indicating that they basically contained all the information of the original impact factors. Subsequently, the three principal components were used to analyse $y$. According to Table 10, the linear equations were obtained as follows:

$$F_1 = 0.3589x_1 - 0.5745x_2 + 0.3797x_3 + 0.2533x_4 + 0.5738x_5 + 0.0598x_6 \tag{19}$$

$$F_2 = 0.4464x_1 + 0.5176x_2 + 0.1273x_3 - 0.0852x_4 + 0.119x_5 + 0.7037x_6 \tag{20}$$

$$F_3 = 0.4448x_1 - 0.0788x_2 - 0.2944x_3 + 0.6988x_4 - 0.4700x_5 - 0.0069x_6 \tag{21}$$

Principal component $F_1$ could be almost interpreted as precipitation on the lake surface area ($x_2$) and precipitation across the entire basin area ($x_5$), principal component $F_2$ as the empirical parameter representing catchment characteristics ($x_6$), and principal component $F_3$ as lake surface evaporation ($x_4$). The *catchment factor* represents the catchment change; the precipitation and lake evaporation factor represents the climate change. Taking $F_1$ (precipitation factor), $F_2$ (catchment factor), and $F_3$ (evaporation factor) as independent variables and $y$ as a dependent variable, multiple linear regression was carried out, and the partial least squares [43] was adopted to obtain the following equation:

$$y = 384.0088 - 0.3556F_1 - 0.3061F_2 + 1.9858F_3 \tag{22}$$

**Table 9.** Eigenvalue of the correlation coefficient matrix and variance contribution rate.

| Principal Components | The Eigenvalue | Contribution Rate (%) | Cumulative Contribution Rate (%) |
|---|---|---|---|
| $F_1$ | 3.6055 | 60.0919 | 60.0919 |
| $F_2$ | 0.9272 | 15.4527 | 75.5446 |
| $F_3$ | 0.8867 | 14.7779 | 90.3225 |
| $F_4$ | 0.3052 | 5.0872 | 95.4097 |
| $F_5$ | 0.249 | 4.1507 | 99.5604 |
| $F_6$ | 0.0264 | 0.4396 | 100 |

**Table 10.** Eigenvectors of the correlation coefficient matrix.

| Principal Components | $x_1$ | $x_2$ | $x_3$ | $x_4$ | $x_5$ | $x_6$ |
|---|---|---|---|---|---|---|
| $F_1$ | 0.3589 | −0.5745 | 0.3797 | 0.2533 | 0.5738 | 0.0598 |
| $F_2$ | 0.4464 | 0.5176 | 0.1273 | −0.0852 | 0.119 | 0.7037 |
| $F_3$ | 0.4448 | −0.0788 | −0.2944 | 0.6988 | −0.47 | −0.0069 |

By converting the influence of principal component factors on underground runoff into percentages, it could be known that evaporation had the largest influence on underground runoff (75%), while precipitation and catchment modifications had a significant influence on the underground runoff (−13.43% and −11.57%, respectively) with an inverse relationship.

Therefore, the contribution rate of climate change to underground runoff can be estimated to be 83.43%, while catchment modifications correspond to −11.57%.

### 3.2.5. Summary

The partial least squares method was used to analyze the contribution rate of $P_l$, $R_{ls}$, $E_l$ and $R_{lg} \pm \varepsilon$ to $\Delta h$ according to Equation (1), and the relative contribution rate obtained was 20.86%, 29.58%, −41.28% and 9.36%, respectively.

The contribution rates of climate change and catchment variability to ($\Delta h$) were obtained by the Equations (3) and (4). It was concluded that the contribution rate to the lake water level variations caused by climate and catchment factors was 93.13%, and 6.87%, respectively (Table 11).

$$R(\Delta h)_c = 20.64\% + 40.84\% + 29.26\% \times 80.19\% + 9.26\% \times 88.44\% = 93.13\% \tag{23}$$

$$R(\Delta h)_u = 29.26\% \times 19.81\% + 9.26\% \times 11.56\% = 6.87\% \tag{24}$$

**Table 11.** Contribution rate of the hydro-climatic factors.

| Contribution Rate | $P_l$ | $R_{ls}$ | $E_l$ | $R_{lg} \pm \varepsilon$ | $\Delta h$ |
|---|---|---|---|---|---|
| Climate Changes | 100 | 80.19 | 100 | 88.44 | 93.13 |
| Catchment Modifications | 0 | 19.81 | 0 | 11.56 | 6.87 |

## 4. Discussion

### 4.1. Relationship between the Hydro-Climatic Factors and Lake Water Level Variations

In line with global warming consequences, high temperatures enhanced water vapor transport and redistribution across the entire catchment area, increasing the precipitation rates on the TP (the precipitation recorded in the basin under investigation increased by 1.4347 mm/year). Furthermore, datasets demonstrated that when temperatures increased, the potential evapotranspiration showed a decreasing trend, which confirmed the theory of the "Evaporation Paradox" [54]. Furthermore, according to datasets recorded, the annual maximum temperature and annual minimum temperatures in the basin had risen during the period of study, while the solar duration and wind speed had

significantly decreased [55,56]; therefore, these factors may have had a crucial impact on the reduction of potential evapotranspiration [57]. The evaporation rate in the lake decreased at the beginning (1960–1967), then increased (1968–1979), then declined again towards the end of the period under investigation (1980–2016). During the period from 1960 to 2004, the main reasons for the decline of water levels were the overall strong evaporation, the lack of rain and runoff, and the decrease of evaporation since 1980 could not reverse this negative trend. On the other hand, since 2005, the water level of the lake increased and this could have been due to increased precipitation recorded, combined with more runoff and lower evaporation rates.

Song et al. pointed out that runoff in most parts of the world has been decreasing significantly [58], such as in southern Australia, southern Europe, the southern region of South America and the western region of North America [59], as well as in most areas in the North of China (such as the Huaihe River [60]). While previous studies completed by Zhao et al. [61] showed that the annual runoff reduction at four main hydrologic stations in the Yellow River basin (a neighborhood area adjacent to the one investigated by this study) ranged from 17.93% to 40.79%, the results of this study showed that runoff in Qinghai Lake basin presented an upward trend, which was similar to the research of Wang et al. [62]. The results of runoff evolution attribution analysis showed that the increase in precipitation and the decrease of evaporation are the main factors leading to the increase in runoff. The trends of surface runoff and water level variations of lake were strongly consistent, and water level variations were largely affected by the effects of the climate factors. The change in precipitation had a more obvious influence on the runoff in the basins of TP, which are relatively arid, than in the humid area.

It was found that fluctuation of the annual underground runoff was not only affected by precipitation, evaporation and infiltration of surface runoff in the lake area and surrounding areas, but was also related to the fluctuation of water levels of the lake [29], and there was a noticeable connection between surface water and groundwater [63,64], showing that the runoff into the lake had positive and negative values. Since 2005, the decrease of evaporation and the increase in precipitation changed the conversion process of surface runoff and underground runoff, and the negative values increased significantly, indicating that more and more water in the lake was replenishing groundwater.

## 4.2. Relationship between the Catchment Modifications and Water Level Variations

In general, water level variations in the lake were the result of combined effects due to climate change and human activities. Among them, direct water intake (e.g., agricultural irrigation and drinking water for livestock) only affected the inflow rates into the lake for 4.8% of the total river discharge [28]; hence, it can be considered a negligible factor. By also developing farming areas and reducing forests (especially with local projects started in 2000), direct water intake dropped even more. Therefore, this paper did not consider the influence of direct water intake on water level changes but mainly focused on the influence of climate change and catchment change on water levels.

In those areas potentially affected by major human activities, the changes in $\omega$ were mainly manifested in land use changes and vegetation changes. It was found that 72% of the total grassland showed significant improvement in Qinghai Lake basin [65]; however, Qinghai lake basin is located at a high altitude and it is affected by a cold climate, and has low population density (4.08 people/km$^2$), so there was little impact due to the land use changes. With the implementation of the returning pasture (farmland) to grass project since 1999 and the comprehensive management project in Qinghai Lake basin since 2008, the vegetation condition had been improved, and changes are reflected in $\omega$ trends.

According to the research conducted by Yuan [66], the annual average ground temperature in Qinghai Lake area increased by a rate of 0.74 °C/10 years. The depth of the annual average maximum permafrost region was then reduced by the rate of 11.7 cm/10 years and the change of permafrost layers [25,67] could definitely change the hydrologic processes under investigation. By becoming smaller, the permafrost area could not contribute consistently as previously to regulate the runoff of the catchment.

*4.3. Uncertainty*

The range of hydrological processes typical of great lakes is inherently uncertain, plus data scarcity adds uncertainty and methodological limitations. Firstly, this study assumed that climate and catchment were independent of each other, but the two factors were interacting in nature [68], and the effects could have also cancelled each other out. Secondly, the mathematical statistics method was used to obtain the contribution rate of climate and catchment modifications to underground runoff, but these methods had some limitations within the assumptions. Finally, the presence of permafrost complicated the investigation of hydrological processes and the characterization of their anomaly behaviors associated with climate warming.

Despite these limitations, the main purpose of this study was to use existing monitoring data to analyze the evolution law of Qinghai Lake level, separating the contribution rate of climate and catchment change to the water level variation, and better guide the future water resource management and rational utilization. From 1960 to 2016, the maximum lake area of Qinghai Lake was 4527.3 km$^2$ (in 1960), while the minimum was 4224.3 km$^2$ (in 2004), with a difference of 303 km$^2$, which is equivalent to the size of Co Nag Lake in China, the highest fresh water lake in the world. Therefore, this study can be very useful as a pilot case to associate with other behaviors recorded in lakes with similar and different conditions.

## 5. Conclusions

This study analyzed the trend of water level variation and hydro-climatic factors in Qinghai Lake Basin from 1960 to 2016 and revealed the main causes affecting the lake water levels. The paper provided a reference base for the development and management of water management in this region and provided important insights that could be applied to other basins.

Conclusions can be summarized as follows:

(1) Qinghai lake experienced severe water level fluctuations in the past 57 years. In period I (1960–2004), the annual water level of the lake declined by 3.46 m at the rate of 7.84 cm/year ($P < 0.001$), while it rose by 1.49 m at the rate of 13.80 cm/year ($P < 0.001$) in period II (2005 to 2016). The variation in water level $\Delta h$ mainly tended to increase during the study period, and the water quantity of the lake increased, passing temporarily from a deficit rate to a surplus one.

(2) The correlation relationships between $E_1$, $P_1$, $R_{ls}$, $R_{lg} \pm \varepsilon$, $\omega$ and $\Delta h$ followed this order: $E_1$ (−0.705) > $R_{ls}$ (0.590) > $P_1$ (0.356) > $\omega$ (−0.262) > $R_{lg} \pm \varepsilon$ (0.143). Overall, the major cause of water level change in Qinghai Lake was the combined effect of evaporation (causing a reduction in water quantities), and precipitation (causing a surface runoff increase).

(3) The contribution rate of multiple factors to the water balance of Qinghai Lake Basin to $\Delta h$ was quantified and it can be classified as follows: $E_1$ (−49.34%) > $P_1$ (29.82%) > $R_{ls}$ (16.76%) > $R_{lg} \pm \varepsilon$ (4.08%). Among all the factors investigated, $E_1$ and $P_1$ belong to climate change factors; hence, by combining the contribution rates of climate change and catchment change induced by human activities to $R_{ls,}$ the results obtained were 80.19%, 19.81%, respectively, and those related to $R_{lg} \pm \varepsilon$ were 8.44%, −11.56%, respectively. Therefore, the contribution rate for both groups of parameters to $\Delta h$ was in total 93.13%, 6.87%, respectively. The results showed that climate change was the leading cause of significant changes in water levels in the lake.

(4) The impact of global climate change on the hydrology and environment of the Tibetan Plateau was clear, strongly confirming the high sensitivity of great lakes on the Tibetan Plateau to climate change, and solutions need to be adopted to enable strategies to deal and cope with future climate change scenarios.

**Author Contributions:** J.F. and G.L. and J.Z. designed the study. J.F. performed the analysis and wrote the first draft of the manuscript. M.R., G.M., X.Y. and H.W. contributed to reviewing and editing the final version of the manuscript, software: G.J.

**Funding:** This research was supported by the National Key Research and Development Program of China (2016YFC0500802), and the Beijing Municipal Education Commission (CEFF-PXM2019_014207_000099), and Special Funds for Scientific Research of Forestry Public Welfare Industry (201404308), and Driving Analysis of Extreme Climate Events on Variation of Runoff and Sediment Discharge in Jinsha River Basin, the National Natural Science Foundation of China (Grant No. 41601279).

**Acknowledgments:** The authors would like to give special thanks to the Institute of Information Center of Qinghai Hydrographic Bureau, China (ICQHB) for providing data on water levels, evaporation and surface runoff.

**Conflicts of Interest:** The authors declare no conflict of interest.

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
