# Peer review of "Analysis of Long-Term Water Level Variations in Qinghai Lake in China"

_water, doi:10.3390/w11102136_

Round 1

Reviewer 1 Report

Abstract 

L16. Do you mean drivers instead of divers

L21-22. 'Authors have' not clear as this is your own work.

Introduction

The first paragraph doesn't flow properly. Needs to be rewritten.

L57-59, Please provide a reference

Reviewer 2 Report

This work is very interesting. The authors investigated the empirical data with notable techniques. The results are meaningful and will provide new knowledge into the community. I have no doubt to accept this good research paper.

Author Response

Many thanks for the positive comments and for the input to improve the manuscript.

Reviewer 3 Report

The article deals with interesting statistical analysis of long-term water level variations in selected lake in China. The standard but quite advanced statistical methods based on trend and correlation analysis were used for the task solution. A sufficient monitoring data for the period of 1960-2016 have been considered and overall the results are very well supported by meteorological, hydrological and land use data.

The Table 1 is not necessary and should be deleted as we have the stations names and locations in the fig. 1 and the data collection period is the same for all of them and is mentioned in the text.

Rls is explained as the yearly surface runoff, would be better to add that it indicates river runoff, because surface runoff stands for a separate balance component given as a direct runoff to the lake. Additionally the surface runoff is not considered in analysis, maybe is negligible, but it should be mentioned.

Why not to change the colour of 5 year moving average to better distinguish from period II in the fig.3? By the way in explanations periods? or rather period I, period II.

The statement in lines 509-511 that underground runoff is affected by such obvious phenomena like precipitation, evaporation or water level variation seems to be unnecessary. Similarly the conclusion (2) lines 568-570 is obvious and somehow trivial considering such an advanced analysis.

Figure 5 is not visible, so expand the size or choose 4 maps only.

A space character is omitted many times between values and units like e.g. in lines: 275-277, 306, 328, 552, 563, table 5...

The brackets for units must be unified like (mm) and [mm] – p. 301

The paper is clearly written, with sufficient explanation of the methods. The problem was solved and the conclusions are properly presented, however some of them are too obvious.

All the references seem to be sufficient and cited in the paper. Describing calculations of water balance components to the lake please consider to complete with A. A. El-Zehairy, M. W. Lubczynski, J. Gurwin, 2018: Interactions of artificial lakes with groundwater applying an integrated MODFLOW solution. Hydrogeol J (2018) 26:109–132.

So, please consider the suggestions given above.
